# Antihyperalgesic and Antiallodynic Effects of Amarisolide A and *Salvia amarissima* Ortega in Experimental Fibromyalgia-Type Pain

**DOI:** 10.3390/metabo13010059

**Published:** 2022-12-30

**Authors:** Gabriel Fernando Moreno-Pérez, María Eva González-Trujano, Alberto Hernandez-Leon, María Guadalupe Valle-Dorado, Alejandro Valdés-Cruz, Noé Alvarado-Vásquez, Eva Aguirre-Hernández, Hermelinda Salgado-Ceballos, Francisco Pellicer

**Affiliations:** 1Laboratorio de Neurofarmacología de Productos Naturales, Dirección de Investigaciones en Neurociencias, Instituto Nacional de Psiquiatría Ramón de la Fuente Muñiz, Calz. México-Xochimilco 101, Col. San Lorenzo Huipulco, Tlalpan, Ciudad de México 14370, Mexico; 2Posgrado en Ciencias Biológicas, Facultad de Medicina, Universidad Nacional Autónoma de México, Ciudad Universitaria, Ciudad de México 04510, Mexico; 3Laboratorio de Neurofisiología del Control y la Regulación, Dirección de Investigaciones en Neurociencias, Instituto Nacional de Psiquiatría Ramón de la Fuente Muñiz, Calz. México-Xochimilco 101, Col. San Lorenzo Huipulco, Tlalpan, Ciudad de México 14370, Mexico; 4Departamento de Bioquímica, Instituto Nacional de Enfermedades Respiratorias Ismael Cosio Villegas, Calz. de Tlalpan 4502, Col. Sección XVI, Tlalpan, Ciudad de México 14080, Mexico; 5Laboratorio de Productos Naturales, Departamento de Ecología y Recursos Naturales, Facultad de Ciencias, Universidad Nacional Autónoma de México, Ciudad Universitaria, Ciudad de México 04510, Mexico; 6Unidad de Investigación Médica en Enfermedades Neurológicas, Hospital de Especialidades, Centro Médico Nacional Siglo XXI, IMSS, Ciudad de México 06720, Mexico

**Keywords:** nociplastic pain, depression, fibromyalgia, neoclerodane diterpene, *Salvia*

## Abstract

*Salvia amarissima* Ortega is an endemic species of Mexico used in folk medicine to alleviate pain and as a nervous tranquilizer. The *S. amarissima* extract and one of its abundant metabolites, identified and isolated through chromatographic techniques, were investigated to obtain scientific evidence of its potential effects to relieve nociplastic pain such as fibromyalgia. Then, the extract and amarisolide A (3–300 mg/kg, i.p.) were pharmacologically evaluated in reserpine-induced fibromyalgia-type chronic pain and in depressive-like behavior (as a common comorbidity) by using the forced swimming test in rats. The 5-HT_1A_ serotonin receptor (selective antagonist WAY100635, 1 mg/kg, i.p.) was explored after the prediction of a chemical interaction using in silico analysis to look for a possible mechanism of action of amarisolide A. Both the extract and amarisolide A produced significant and dose-dependent antihyperalgesic and antiallodynic effects in rats, as well as significant antidepressive behavior without sedative effects when the antinociceptive dosages were used. The 5-HT_1A_ serotonin receptor participation was predicted by the in silico descriptors and was corroborated in the presence of WAY100635. In conclusion, *S. amarissima* possesses antihyperalgesic, antiallodynic, and anti-depressive activities, partially due to the presence of amarisolide A, which involves the 5-HT_1A_ serotonin receptor. This pharmacological evidence suggests that *S. amarissima* and amarisolide A are both potential alternatives to relieve pain-like fibromyalgia.

## 1. Introduction

According to the International Association for the Study of Pain (IASP), “pain” is defined as an upsetting sensory and emotional experience related to or resembling actual or potential tissue damage [1]. Chronic pain is a typical, complex, and upsetting issue that profoundly affects people and society [2] and influences 25 to 29% of the population, although a few reports have revealed a commonness range from 8 to 80%. Those disparities are due to methodological discussions concerning the meaning of “chronicity” [3].

The international community of pain researchers has suggested the term “nociplastic pain” to describe a third category of pain that underlies mechanisms of action incompletely understood that includes fibromyalgia syndrome (FM) [4]. FM is considered a disorder of chronic generalized muscular pain, stiffness, generalized fatigue, sleep abnormalities and cognitive problems, mainly associated with depression [5,6]. It affects about 5% of adults, mainly women 20–60 years old, and is exacerbated at menopause. It is characterized by allodynia, which is pain derived from an otherwise non-painful stimulus [7], and hyperalgesia [8], defined as an exaggerated response to a painful stimulus [9]. In addition, people with FM have reduced tolerance to pain, especially at extremes of heat and cold [10]. There is evidence of an altered circuity in the pain networks and abnormal processing of pain in FM [8,11]. However, the pathogenesis of this disease remains unknown, including missing objective diagnostic criteria and a complete and effective therapy [5,7,12].

Current pharmacological therapy for FM includes one or more drugs acting at the central level and selected based on their proven efficacy as pain modulators, such as tricyclic antidepressant drugs or serotonin–norepinephrine reuptake inhibitors, as well as gabapentinoids and other membrane stabilizers, including herbal remedies [13,14,15]. Non-steroidal anti-inflammatory drugs are not effective for this type of pain, and opioid drugs are avoided because of their association with a high rate of multiple adverse effects and development of tolerance to analgesic effects as well as physical and psychological dependence [15]. Although tramadol produces analgesic effects by modulation of the *mu*-opioid receptors, it also acts through atypical serotonin and noradrenaline reuptake inhibition, producing a moderate efficacy for FM pain, alone and in combination with antidepressants or other analgesic drugs [15]. Preclinical studies have reported the antihyperalgesic and antiallodynic effects of tramadol (TR) used as a reference drug in reserpine (RES)-induced FM in rats [16,17]. Moderate analgesia was reported in patients with severe FM when TR was used in combination with paracetamol [18,19]. Therefore, finding effective and safe therapeutic alternatives for treating the FM condition is a continuous and important challenge to increase the quality of life in patients.

Medicinal plants are not only a potential resource for pain therapy per se but are also a source of raw material to obtain new molecules of pharmacological interest [20]. For example, terpenoids are biomolecules that have demonstrated an interesting pharmacological response with a moderate dosage range and few possible adverse effects in the treatment of pain [21,22,23,24]. Many *Salvia* species are used for ethnomedical purposes to alleviate a wide variety of ailments [23,25]. The reported pharmacological properties of this genus are relevant effects such as anti-inflammatory, anti-viral, antiprotozoal, cytotoxic and phytotoxic agents [26,27]. *Salvia* species constitute a rich source of terpenoids, mainly neoclerodane diterpenes, which have been shown to produce antinociceptive activity in several experimental models [23,25,28,29] where nociceptive, visceral, and abdominal pain were explored [22,24,30,31,32], or even neuropathic pain in rodents [21]. *S. amarissima* has been evaluated to be prepared as an extract of different polarity, where amarisolide A was identified as one of the most abundant terpenes with a chemical structure of neoclerodane [22,33]. However, the potential antinociceptive effects of *S. amarissima* extracts or amarisolide A have not yet been explored in FM-type pain in order to provide evidence of a broader spectrum of pain-relieving biological activity of this medicinal plant and one of its abundant bioactive metabolites. In the present study, the reserpine (RES)-induced experimental FM model was used to investigate the antiallodynic and antihyperalgesic effects of *S. amarissima* by administrating both several doses of a medium polar extract containing an abundant presence of amarisolide A and the independently purified metabolite. In addition, a possible mechanism of action was investigated through receptor antagonists supported by an in silico analysis, as well as the antidepressant activity, since depression is a comorbidity associated with FM.

## 2. Materials and Methods

### 2.1. Plant Material

The aerial parts of *S. amarissima* Ortega were collected in August 2018 in the community of Santiago Huauclilla, Oaxaca, Mexico. One specimen was authenticated by the expert taxonomist Dr. Martha Juana Martínez-Gordillo and deposited at the IMSS Herbarium in Mexico City, Mexico, with voucher number 16360.

The plant material was dried at room temperature and then finely ground. A 50 g sample of pulverized vegetal material was processed by maceration in organic solvents of increasing polarity (150 mL) to prepare five extracts: hexane (HEX), dichloromethane (DCM), ethyl acetate (EtOAc), acetone (AC), and methanol (MeOH) with the following yields: HEX, 0.26 g (0.52%); DCM, 3.63 g (7.26%); EtOAc, 2.31 g (4.62%); AC, 2.22 g (4.44%); and MeOH, 4.88 g (9.76%). After 72 h maceration of exposure to the solvent, each extract was obtained and then subjected to rotatory evaporation in a R-210 equipment (Büchi, Flawil, Switzerland) three consecutive times, filtering it and separating it from each residue. In addition, an aqueous extract was prepared with 50 g of raw material in decoction for 20 min, filtered, and lyophilized to obtain 5.125 g (10.25%) in order to analyze the concentration of amarisolide A in comparison with the organic extracts.

The presence of terpenes in the extracts was compared by thin layer chromatography (TLC), which was revealed with anisaldehyde sulfuric acid as a spray reagent under ultra-violet light (365 nm) [29]. The EtOAc extract was recognized as the one with the highest abundances of amarisolide A; thus, a second batch of vegetal material (540 g) was processed to obtain more EtOAc extract (25 g, 4.62%) and to subsequently purify amarisolide A. This second extract was fractionated on open column chromatography (OCC) to separate 250 sub-fractions (20 mL each) by using solvents of increasing polarity (HEX, DCM, and AC). These subfractions were grouped into 8 pools according to similarity on TLC plates (3.5 × 7 cm) using EtOAc:MeOH (95:5) as the mobile phase. The presence of amarisolide A was detected as an intense purple spot after anisaldehyde sulfuric acid spray reagent. The highest concentration of the compound was observed in the subfractions obtained with AC in the OCC. Amarisolide A was isolated as a precipitate and then purified by recrystallization as yellow crystals in a yield of 1.2575 g (5.03%) from the EtOAC extract (25 g), finally compared with the preliminarily [22,33], which was used as a standard for HPLC analysis.

### 2.2. High-Performance Liquid Chromatography (HPLC) Analysis

The chromatographic profile of the EtOAc extract was performed by using samples of 5 mg/mL and amarisolide A at 1 mg/mL (all samples were dissolved in HPLC grade MeOH). After filtration through a 0.22 μm filter (GHP filters, Acrodisc 13, Waters), samples were injected into a UPLC Waters Acquity-H class equipment fitted with a photodiode array detector (UPLC, Acquity Waters, Wexford, Ireland). Data were processed with Empower chromatographic software version 3 (Waters, Milford, MA, USA) under the following conditions: 10 μL injection volume; Column: Symmetry C-18 (150 × 4.6 mm, 5 mm, 100 A (Waters, Wexford, Ireland) with thermostat at 35 °C. The mobile phase consisted of acetonitrile/water acidified with 0.1% phosphoric acid, v/v. The initial gradient mixture was 70% A and 30% B. The concentration of solvent B was gradually increased to 70%. Finally, gradient returned to the initial concentrations (70%:30%) over a total time of 12 min at a constant flow rate of 0.7 mL/min and a detection wavelength of 254 nm [33]. Amarisolide A was detected at the peak found at 1.9 min of retention time in the chromatographic profile.

### 2.3. Animals

Male Wistar rats (200–230 g) were used in groups of at least 6 animals. Animals were kept in acrylic boxes with access to water and food ad libitum in a temperature-controlled room (22 ± 2°C) and light under a 12 h light/dark cycle (light on at 7:00 a.m.) until experiments began. The guidelines established by the bioethics committee of the local institution (CONBIOETICA-09-CEI-01-20170316), as well as national (NOM-062-ZOO-1999) and international guidelines for the care and use of laboratory animals were followed. This research was approved by the research committee of the Instituto Nacional de Psiquiatría Ramón de la Fuente Muñiz in research protocols no. NC123280.0 and NC17073.0 dated August 2012 and October 2017, respectively.

### 2.4. Drugs and Reagents

Tramadol (TR, reference drug) was obtained from Laboratories Amsa S.A de C.V (Antibióticos de México, S.A. de C.V. AMSA, Mexico). WAY100635, Reserpine (RES), acetic acid, and Tween 80 were purchased from Sigma-Aldrich (St. Louis, MO, USA). IL-1beta anti-inflammatory cytokine kits were purchased from Thermo Fisher Scientific (Waltham, MA, USA). RES was dissolved in its vehicle (0.5% acetic acid) for subcutaneous (s.c.) administration. The organic extract and amarisolide A were resuspended in 0.2% tween 80 in distilled water. The vehicle group received 0.2% tween 80 in the same volume and the same route of administration. The reference drugs and treatments to explore the mechanism of action were dissolved in distilled water. Drugs were freshly prepared on the day of the experiments and administered intraperitoneally (i.p.) in a volume of 1 mL/kg. Doses were selected from previous reports of literature or preliminary experiments in our laboratory [22,33].

### 2.5. Pharmacological Evaluation

#### 2.5.1. Experimental Design

The design included eleven groups with at least eight rats in experimental groups (1 to 8, 10, 12, and 13) and six rats in the reference drug (9) and naïve groups (11). The acute (single dose) treatments administered were as follow: groups 1–4: EtOAc extract of *S. amarissima* (ESA) (3, 30, 100, and 300 mg/kg, i.p.); groups 5–8: amarisolide A (3, 10, 100, and 300 mg/kg, i.p.); group 9: positive control drug TR (10 mg/kg, i.p.); group 10: FM (RES) i.e., rats receiving the vehicle, and group 11: naïve group. The administration of treatments was carried out on the fifth-day post-induction when maximum nociception was detected using the hyperalgesic and allodynic threshold values [34,35]. All treatments were evaluated in a temporal course for a period of 4 h with measurements at 0, 30, 60, 120, 150, 180, and 240 min after administration. The data from the temporal course curves were converted to the area under the curve (AUC) to build dose–response plots and determine efficacy, potency, and tolerated doses of treatments. 

Other groups were used to explore a possible mechanism of action involved in the antihyperalgesic and antiallodynic effects of amarisolide A: group 12, where the serotonin 5-HT_1A_ receptor was assessed by using the selective antagonist (WAY100635, 1 mg/kg, i.p.) alone, and group 13: in which rats received the antagonist 15 min before amarisolide A (100 mg/kg, i.p.) (see timeline in Figure 1).

Since depression is closely associated with FM, and antidepressant drugs are currently considered one of the most effective therapies, independent groups were used to assess the antidepressant activity of ESA at doses of 3, 10, and 30 mg/kg (groups 14–16) and amarisolide A at doses of 10 and 30 mg/kg, i.p. (groups 17–18) compared to the antidepressant effects of the reference drug fluoxetine (group 19) using three typical doses already reported in the literature [36,37].

#### 2.5.2. Induction of Experimental FM-Type Pain

Experimental FM-type pain was induced using the method previously described by Hernandez-Leon et al. [35] modified from Nagakura et al. [34]. Briefly, animals were habituated for three days before induction to the experimental conditions for measurements of hyperalgesia and allodynia. Then, always in the neck of rats, RES (1 mg/kg, s.c.) was administered once per day for 3 consecutive days in a volume of 1 mL/kg. On the day of the experiments, animals were individually placed in acrylic cylinders to habituate for 30 min prior to the assessment of nociceptive behavioral response hyperalgesia and allodynia.

##### Mechanical Hyperalgesia (Muscular Pressure Threshold)

Tenderness to palpation is one of the diagnostic criteria for FM [38], where application pressure is routinely used to evaluate muscular pain thresholds in humans with FM [39]. In this study, the muscle pressure threshold was recorded in rats using the Randall–Sellito apparatus (Ugo-Basile, Varese, Italy) as previously reported by Nagakura et al. and Hernandez-Leon et al. [34,35]. For this evaluation, rats were immobilized using a cloth while exposing the right hind paw to apply increasing linear pressure on the medial gastrocnemius muscle (250 g maximum force). The behavioral response was considered positive when rats withdrew their limbs or vocalized. The mean of three trials was calculated for each session, with an interstimulus interval of at least 1 min.

##### Tactile Allodynia (Tactile Response Threshold)

Allodynia is one of the main clinical characteristics that is correlated in patients with FM [38]. In experimental FM, the tactile response threshold was recorded using the von Frey filaments (Stoelting Co., Wood Dale, IL, USA) with the up and down method [40]. This evaluation consisted of placing each rat into a transparent cage with a metal mesh floor and allowing for habituation for at least 30 min. Then, a touch with a von Frey filament (2, 4, 6, 8, 10, and 15 g) was applied to the plantar surface of the right hind paw increasing the caliber. Paw withdrawal due to filament pressure was defined as a positive response, whereas no paw withdrawal within 6 s was defined as a negative response. The test started with the application of the 2 g force filament. The next filament was smaller or higher depending on the occurrence of the positive or negative response. The test ended when four responses were obtained after the first filament change. The paw withdrawal threshold was calculated as the tactile response threshold using an adaptation of the Dixon’s up–down paradigm [40].

##### Cold Allodynia (Threshold for The Cold Stimulus)

Thermal allodynia was tested through cold stimuli using the acetone test [41]. For this assay, each rat was individually placed in a transparent compartment on a raised metal grid to allow for habituation for at least 30 min. Then, a volume of 50 μL of acetone was sprayed on the middle part of the dorsal surface of the right hind paw of the rat using a drip adapter on an insulin syringe (Becton Dickinson de México, S.A de C.V. Mexico City). Immediately after the spray and for a period of 60 s, the response time in which the rat licked, shook, flinched, or raised the right hind paw was counted. This test was performed three times with an interval of 5 min between each measurement, and the average was considered as the cold allodynia threshold [42].

#### 2.5.3. Mechanism of Action of Amarisolide A

In the analgesic effects of natural products such as terpenes, the participation of the serotonin 5-HT_1A_ receptor has been reported [24,43], not only in visceral or inflammatory pain but also in neuropathic pain [44,45,46]. To explore the involvement of this receptor as a possible mechanism of action in the antinociceptive effect of amarisolide A, the serotonin 5-HT_1A_ receptor antagonist WAY100635 (1 mg/kg, i.p.) was administered 15 min before treatment with amarisolide A (100 mg/kg, i.p.).

At the end of the experiments, rats were euthanized by decapitation to dissect the cortical and hippocampal. Brain regions as well as the thoracic and lumbar spinal cord regions were divided into dorsal and ventral sections for subsequent cytokine analysis. 

#### 2.5.4. In Silico Analysis of Amarisolide A

An in silico analysis was performed to assess whether there is a role of serotonin 5-HT_1A_ inhibitory receptors in producing the antihyperalgesic and antiallodynic effects of amarisolide A. The crystal structure of the compound amarisolide A was obtained from the PubChem database. The structure was protonated using the Avogadro software V1.2.0 (Avogadro Chemistry, Pittsburgh, PA, USA) at pH 7.4, and the minimum energy spatial configuration was subsequently determined using the Merck Molecular Force Field (MMFF94, Merck Research Laboratories, Boston, MA, USA). The protein structure of serotonin 5-HT_1A_ receptor (7E2X) was obtained from the Protein Data Bank (PDB, https://www.rcsb.org/ (accessed on 14 November 2022)), selecting a resolution of less than 3 Å. Then, the docking was performed with the CB-Dock tool [47]. The results of the CB-Dock tool software V2.0 (Structural Bioinformatics Research group, Chengdu, China) were contrasted with UCSF Chimera 1.16 (Resource for Biocomputing, Visualization, and Informatics University of California, San Francisco, CA, USA) to protein preparation and Autodock vina 1.1.2 (Oleg Trott, La Jolla, CA, USA).

#### 2.5.5. Antidepressant Activity of the *S. amarissima* EtOAc Extract and Amarisolide A 

In this study, the modified rat forced swimming test (FST) [48,49] was performed in rats. Briefly, rats were individually placed in a glass cylinder (46 cm height, 20 cm diameter) filled with 30 cm of tap water (25 ± 2 °C). The FST consisted of two swimming sessions. In the first session (pre-test), rats swam for 15 min. Twenty-four hours later, the animals were subjected to a second 5 min session (test). At the end of each session, rats were removed from the cylinder, patted dry with a paper towel, and placed in a cage for 15 min before being returned to their home cages. The test was videotaped and scored by an observer blinded to the treatment.

A time-sampling technique was used to score the presence of the following behaviors: (a) immobility, floating without struggling and making only small movements necessary to keep the head above the surface of the water; (b) swimming, active movements, i.e., animals that move and dive around the cylinder; or (c) climbing, rats that make vigorous movements with their forepaws in and out the water, generally directed toward the wall of the cylinder. Results were expressed as the mean number of behavior counts ± S.E.M every 5 s for 5 min.

Fluoxetine (10 mg/kg, s.c.) was used in a subacute administration by applying three individual injections (10 mg/mL) into the neck of the rat in a volume of 1 mL/kg body weight. The first and second injections were given 23 and 5 h before the experiment, respectively, and a further injection was given 1 h prior to the behavioral evaluation. This dose regimen is already reported in literature [36,37].

##### Ambulatory Activity

The rats were individually placed inside a cage (75 × 68 × 65 cm) divided into 9 squares. The number of squares explored by the rat during 2 min was registered [50,51]. A square was considered explored when all four paws of the rat were within the square. A significant decrease in the number of explored squares was considered a sedative effect. The field was wiped with a cleaning solution and completely dried before it was used by the next rat. 

#### 2.5.6. Determination of Inflammatory Cytokines in the Nervous Tissue of Rats with FM

IL-1β levels present in homogenates of the prefrontal cortex, hippocampus, and spinal cord divided into thoraco-dorsal, thoraco-ventral sections, lumbo-dorsal, and lumbo-ventral nervous tissue of FM rats were measured by enzyme-linked immunosorbent assay (ELISA) using commercial kits according to the manufacturer’s instructions (ELISA kit for rats, Thermo Fisher Scientific, Waltham, MA, USA). Assays were carried out in duplicate, and the detection limits for IL-1β were 5.52 pg/mL. Data were collected and analyzed using an Agilent BioTek Epoch Spectrometry Microplate Reading System (Santa Clara, CA, USA) to calculate IL-1β concentrations in the sample from standard curves.

### 2.6. Statistical Analysis

Data are expressed as the mean ± S.E.M of at least six or eight replicates depending on the treatment (control or experimental). Temporal course curves (TCC) data were examined by repeated measures in a two-way analysis of variance (ANOVA) followed by Dunnett’s post hoc test. The area under the curve (AUC) was calculated using the trapezoidal method to build the dose–response plot whose data were analyzed by a one-way ANOVA followed by the Dunnett’s post hoc test for comparison vs. vehicle group, or Tukey’s test for multiple comparison. Statistical analysis was performed using GraphPad Prism software version 8.0.2 for Windows (GraphPad Software INC, La Jolla, CA, USA). *p* < 0.05 was considered to declare a significant difference.

## 3. Results

### 3.1. The Presence of Amarisolide A Presence in the Extract of S. amarissima

The amarisolide A compound was detected as one of the most abundant metabolites in *S. amarissima* distributed in several organic extracts but mainly concentrated in the EtOAc extract. HPLC chromatographic profiles allowed for the identification of this neoclerodane terpene (5 mg/g de extract) at a retention time of 1.99 min with a wavelength of 256 nm (Figure 2).

### 3.2. Antihyperalgesic and Antiallodynic Effects of S. amarissima EtOAc Extract and Amarisolide A

The muscle pressure threshold in the RES group was significantly decreased (58%) on the 5th day post-induction, thus stablishing the FM-type pain condition at 114.10 ± 9.38 g compared to the observed response in the naïve group (238.8 ± 7.58 g), as shown in the TCCs (Figure 3A). With the cold stimulus, in the RES group, the time that the rat spent flinching or licking the right paw (3.90 ± 0.6 s) was significantly increased by almost seven-fold more than in the naïve group (0.49 ± 0.2 s) (Figure 3C). Regarding the tactile response in FM rats, a decrease (69.24%) in the withdrawal threshold was observed, from 14 ± 2 g in the naïve group vs. 4 ± 2 s in the RES group (Figure 3E). 

The ESA (30–300 mg/kg) produced a significant antihyperalgesic effect between 30 and 60 min after its administration. The antinociceptive response of the reference drug (TR, 10 mg/kg) was significant from the first 30 min, reaching the response of the naïve group, but it only lasted 120 min (Figure 3A). In contrast, the significant effects of ESA remained until the end of the experiment (240 min) (Figure 3A; treatment: F_6,44_ = 13.78, *p* < 0.001; time: F_6.217, 273.5_ = 5.948, *p* < 0.001; interaction: F_48,352_ = 7.488, *p* < 0.001). A dose-dependent response was observed in the presence of ESA when AUC was calculated from TCCs, producing a significant effect of 27.65% (3 mg/kg), 36.44% (30 mg/kg) and 51.81% (100 or 300 mg/kg), resembling the effect of the reference drug (TR, 10 mg/kg) (F_6,44_ = 12.73, *p* < 0.0001). In this way, an effective dose of 50 (ED_50_) = 91.04 mg/kg was calculated for ESA (Figure 3B).

The antiallodynic effect of ESA after cold (Figure 3C) and tactile (Figure 3E) stimuli was observed from the first 30 min of evaluation with all the doses evaluated, except for 3 mg/kg. The significant response was maintained throughout the evaluation period (Figure 3C,E; treatment: F_6,43_ = 48.55, *p* < 0.001; time: F_4.553,195.8_ = 7.81, *p* < 0.001; interaction: F_48,344_ = 2.89, *p* < 0.0001). The AUC_0–240 min_ obtained from TCCs demonstrated a significant inhibition from 0% to 88% of flinching/licking behavior with an ED_50_ = 18.12 mg/kg (F_6,44_ = 52.12, *p* < 0.0001) (Figure 3D). Regarding the withdrawal threshold, a significant and dose-dependent increase was observed in the AUC_0–240 min_ from 0% to 77.87% with an ED_50_ = 58.56 mg/kg (F_6,44_ = 6.27, *p* < 0.0001) (Figure 3F). Both maximum effects were similar to those obtained in the presence of the reference drug (TR, 10 mg/kg) (Figure 3D,F; treatment: F_6,44_ =15.60, *p* < 0.0001; time: F_6.422,282.6_ = 5.97, *p* < 0.0001, interaction: F_48,352_ = 1.56, *p* < 0.0136).

Amarisolide A was the major metabolite present in the EtOAc extract of *S. amarissima* as corroborated by HPLC analysis (Figure 2). Per what was observed in the TCCs, its participation as an antihyperalgesic bioactive compound and as an antiallodynic one in the presence of both a thermal and a mechanical stimulus was confirmed (Figure 4A,C,E). The antinociceptive response was observed from the first 30 min, and all doses were significant, reaching a maximum inhibitory effect with a dosage of 300 mg/kg compared to the rats in the RES group (Figure 4A,C,E; treatment: F_6,42_ = 38.81, *p* < 0.001; time: F_5.436,228.3_ = 3.69, *p* < 0.0023, interaction: F_48,336_ = 4.24, *p* < 0.0001). The AUC_0–240 min_ demonstrated a significant antihyperalgesic effect for amarisolide A starting at 10 mg/kg (18.59%), which increased in the dose of 100 mg/kg (45.09%) or 300 mg/kg (62.96%), calculating an ED_50_ = 201.84 mg/kg (F_6,42_ = 37.52, *p* < 0.0001) (Figure 4B).

In the case of allodynia with tactile, cold, and mechanical stimuli, amarisolide A inhibited nociception at different doses from 10 to 300 mg/kg. In contrast to TR (10 mg/kg), the effects of amarisolide A remained significant throughout the evaluation period (Figure 4D,F, respectively). The allodynic threshold was recovered to reach the levels of the naïve group at some points of the TCCs for both thermal allodynia (treatment: F_6, 41_ = 21.10, *p* < 0.0001; time: F_4.903,201.0_ = 10.08, *p* < 0.001; interaction: F _48, 328_= 2.68, *p* < 0.0001) as for mechanical allodynia (treatment: F_6,41_ = 23.81; *p* < 0.0001, time: F_6.432,263.7_ = 7.17; *p* < 0.001; interaction: F_48,328_ = 2.48, *p* < 0.001) (Figure 4C,E). The AUC_0–240 min_ demonstrated a mechanical antiallodynic effect of 75.97% at 100 mg/kg, where the effect with 300 mg/kg was 47.66% (F_6,42_ = 1.68, *p* < 0.0001), producing an equivalent inhibition to that of TR (10 mg/kg) (57%) (Figure 4D). In the case of thermal allodynia, nociception was inhibited in 3.05%, 42.30%, 83.82%, and 88.29% at doses of 3, 10, 100, and 300 mg/kg, respectively, with an ED_50_ of 12.29 mg/kg (F_6,42_ = 1.89, *p* < 0.0001). TR produced 57% of antinociceptive response (Figure 4F).

### 3.3. Mechanism of Action of Amarisolide A

To explore for a possible mechanism of action involved in the antihyperalgesic and antiallodynic effects of the bioactive metabolite identified in *S. amarissima*. The serotonin 5-HT_1A_ receptor antagonist WAY100635 was administered at a dose of 1 mg/kg, i.p. 15 min before treatment with 100 mg/kg, i.p. of amarisolide A.

When the animals were previously treated with the 5-HT_1A_ receptor antagonist, no antihyperalgesic effect produced by amarisolide A (100 mg/kg) was observed throughout the entire time of the trial (treatment: F_4,29_ = 72.06, *p* < 0.0001; time: F_5,652,163,9_ = 0.9974, *p* < 0.434; interaction: F_32,232_ = 2.53, *p* < 0.0001) (Figure 5A). In the AUC plot, significant differences were observed between the amarisolide A (100 mg/kg) and amarisolide A + WAY100635 groups (F_4,29_ = 62,96, *p* < 0.0001) (Figure 5B). In the presence of the thermal stimulus, the antiallodynic effect of amarisolide A was also prevented from 30 to 240 min of the trial when the antagonist was administrated (treatment: F_4,28_ = 128.10, *p* < 0.001; time: F_5.034,140.9_ = 2.65, *p* < 0.025; interaction: F_32,224_ = 1.85, *p* < 0.0055) (Figure 5C). The AUC plot showed a significant increase in nociceptive behavior in the 5-HT_1A_ receptor antagonist treated group (F_4,29_ = 131,6 *p* < 0.0001). Tactile allodynia increased in the group treated with the antagonist compared to amarisolide A (100 mg/kg) throughout the 240 min of the trial (treatment: F_4,28_ = 88.87, *p* < 0.0001; time: F_5.715,160_ = 3.15, *p* < 0.0068; interaction: F_32,224_ = 2.23, *p* < 0.0004) (Figure 5E). The AUC plot showed a significant decrease in the antiallodynic effect of amarisolide A (100 mg/kg) when animals were previously treated with the 5-HT_1A_ receptor antagonist (F_4,29_ = 89.85, *p* < 0.0001) (Figure 5F).

### 3.4. Molecular Docking

A molecular docking approach was used in the in silico evaluation. The target validation throughout the in silico analysis contributed to a theoretical explanation of the drug–receptor relationship and corroborated the evidence obtained in the in vivo protocol. The results showed that the range of binding energy values for amarisolide A at the 5-HT_1A_ receptor is from −8.1 to −7.2 kcal/mol. The best location of the bonded protein occurs at −8.1 kcal/mol and has 22 amino acid contacts. This result reinforces the in vivo data that demonstrates that the drug–receptor interaction is sufficient to produce analgesia and an antidepressant-like effect. The binding site for amarisolide A is similar to the binding site of an agonist known as buspirone, the docking analysis of which was carried out as a positive control. This agonist produced binding affinities ranging from −7.9 to −6.4 kcal/mol upon contact with 18 amino acids. Buspirone (Figure 6A) and amarisolide A (Figure 6B) docked at sites proximal to the 5-HT_1A_ receptor, sharing 11 docking positions at TYR96, PHE112, ASP116, CYS187, THR188, ILE189, LYS191, PHE361, ALA365, ASN386 and TYR390.

### 3.5. Ambulatory Activity and Forced Swimming Test (FST)

Ambulatory activity evaluated as the number of explored squares showed no significant difference among treatments, including the reference drug fluoxetine or the naïve group (Figure 7A). On the other hand, the results of the FST showed that ESA (10 and 30 mg/kg) and amarisolide A (10 and 30 mg/kg) significantly reduced immobility behavior (F_6,54_ = 4.83, *p* < 0.0005). This reduction was similar to that produced by the reference drug fluoxetine (Figure 7B). ESA (10 and 30 mg/kg) and amarisolide A (10 mg/kg) increased swimming time compared to the naïve group (F_6,54_= 6.95, *p* < 0.0001) (Figure 7C). Both ESA (10 mg/kg) and amarisolide A (30 mg/kg) increased climbing behavior compared to the naïve group (F_6,54_ = 4.80, *p* < 0.0005) (Figure 7D). In the case of fluoxetine, only swimming time increased but not climbing behavior. 

### 3.6. Determination of the Concentration of IL-1β in the Nervous Tissue of Rats in the Reserpine-Induced FM Model

The level of IL-1β (proinflammatory cytokine) in the nervous tissue of rats was quantified by ELISA. Table 1 shows the levels of IL-1β in animals with RES-induced FM. This condition showed an increase in the concentration of this cytokine in the prefrontal cortex (4-fold, F_2,6_ = 3.51, *p* < 0.05), in the hippocampus (1.93-fold, F_2,6_ = 3.64, *p* < 0.01), and in the different sections of the spinal cord: thoraco-dorsal (1.96-fold, F_5,8_ = 2.56, *p* < 0.04), and lumbar-dorsal (2.9-fold, F_2,14_ = 11.90, *p* < 0.05). Treatment with amarisolide A (100 mg/kg, i.p.) reduced the expression of this inflammatory cytokine in the hippocampus (91 %) and in the spinal cord sections in FM rats where significant decreases were observed in lumbar-dorsal (61 %), thoraco-ventral (90%) and lumbar-ventral (72%) sections in comparison with the cytokine level observed in the FM group.

## 4. Discussion

The antinociceptive effects of aerial parts of *S. amarissima*, prepared as aqueous and organic extracts of different polarity, have already been reported by using acute pain models in mice, such as in abdominal [22], nociceptive and inflammatory [33]. The central activity in the antinociceptive effects has been observed not only with the most active extract, ethyl acetate, but also with the abundant metabolite described as amarisolide A [22,33]. These antecedents allowed *S. amarissima* to be considered for its evaluation in an experimental model of nociplastic pain such as that of FM, and thus, they complement the spectrum of antinociceptive activity of this species. In this study, the tolerated doses of the extract, the involvement of 5HT_1A_ receptors as a possible mechanism of action through a prediction of molecular docking and in the presence of an antagonist, as well as the influence of the IL-1β were evaluated.

FM is a chronic and generalized syndrome whose current therapy is not completely effective and safe. In our present study, we evaluated the potential antinociceptive properties of a medium polar extract of *S. amarissima* and its neoclerodane diterpene metabolite, amarisolide A, using the experimental model of FM induced by reserpine administration in rats [34,35]. Firstly, using conventional chromatographic techniques, amarisolide A was isolated and purified from the bioactive extract, obtaining a yielding of 5.03%, which is a higher amount than that reported by Flores-Bocanegra [52], who reported a yield of 2.58% from a DCM:MeOH extract (1:1), while Salinas-Arellano [53] reported 11.6% from an EtOAc partition from an aqueous extract. This information reinforces that differences in the polarity of the solvent, the preparation, the location, and the collection conditions of the plant material are factors that influence not only the yield of the extracts but also the presence of pure metabolites, as in the case of amarisolide A [54].

Concerning amarisolide A, this metabolite belongs to diterpenes of the clerodane skeleton, which are abundant in Mexican *Salvia* species [55,56]. Current research on *Salvia* has reported its antinociceptive efficacy in experimental models of nociceptive and inflammatory pain, where terpenes such as amarisolide A from *S. amarissima* [22,33] and 7-keto-neoclerodan-3,13-dien18,19:15,16-diolide from *S. semiatrata* inhibited abdominal stretching induced with 1% acetic acid, showing a non-dose-dependent response in the first case [22] and in a dependent manner (ED_50_ = 4.15 mg/kg) in the second case [23]. *S. amarissima* has also been reported as an important anti-inflammatory [33]. Other antinociceptive terpenes are salvinorin A and B from *S. divinorum*, which in a dose-dependent manner, also reduced abdominal stretching and time spent licking in the formalin test [24], complemented by antinociceptive effects in inflammatory and neuropathic pain [21]. This information together supports the beneficial properties against painful conditions of some terpenic chemical compounds of *Salvia* species.

The antinociceptive research of *Salvia* has offered the opportunity to explore the possible mechanisms of action of its metabolites as new molecules and drugs for the treatment of pain [24,33]. The efficacy and potency of these metabolites is comparable or better, respectively, to that observed and reported regarding common metabolites found in several species, from different botanical genus and families associated with terpenic nature, for example, β-amyrin, carnosol, oleanolic acid, and ursolic acid, which have already been reported in different models of pain [29,57,58,59,60,61].

The possible nervous activity at the central level of terpenes could be associated with their liposolubility [62], which means that reduced doses are required to produce their pharmacological responses, as in the case of those constituents isolated from medium polar extracts of *S. divinorum* and *S. amarissima*. It is also the case of significant antinociceptive treatment in the mouse formalin test reported with ursolic and oleanolic acids (2.1 and 1.6 mg/kg) [63,64,65]; carnosol (0.5, 1, and 2 mg/kg) [66], and betulinic acid (0.1, 0.5, and 1 mg/kg) [67]. In the present study, evidence was obtained for the first time that medium polarity extract of *S. amarissima* aerial parts produced significant antihyperalgesic and antiallodynic activities, in part due to the major presence of amarisolide A, suggesting potential antinociceptive pharmacological properties of both the plant material as well as this metabolite for the treatment of FM-type pain. Hyperalgesia and allodynia are two painful behaviors associated with prominent stimulation at the CNS level, both observed in the same way as in patients with this syndrome [68]. To the best of our knowledge, there are currently no reports on the efficacy of extracts or pure compounds from the genus *Salvia* in experimental FM. However, some species have been studied in models of nociplastic pain, such as the diterpene tanshinone IIA present in *S. miltiorrhiza*, which in a dose-dependent manner (10, 25, and 50 mg/kg), decreased mechanical hyperalgesia using the rat spinal nerve ligation model [69]. In addition, the diterpene salvinorin A from *S. divinorum* reduced neuropathic-type pain when evaluated in the rat sciatic nerve ligation model [21,70].

A search conducted in Science Direct using the terms “*Salvia*” and “fibromyalgia” returned 12 reports. The terms “diterpene” and “fibromyalgia” in the PubMed repository showed six results. The scarce information on the study of both the *Salvia* genus and terpenoid metabolites as a complementary therapy for the treatment of FM highlights the results found in this study.

Antihyperalgesic and antiallodynic properties of some plant species and their metabolites, as well as their possible mechanisms of action, have been scarcely explored. Some reports have referred to this as in the case of the ethanolic extract (100 mg/kg, p.o.) of *S. lachnostachys* whose antihyperalgesic activity was attributed to the presence of the diterpene fruticulina A (3 mg/kg) by inhibiting TNF-α [71]. The antihyperalgesic effect of the EtOAc extract of *S. divinorum* (100 mg/kg) and salvinorin A (30 to 200 mg/kg, i.p.) in the sciatic nerve constriction neuropathy test was prevented in the presence of the Kappa opioid receptor antagonist nor-binaltorphimine (1 mg/kg) [21]. The ethanolic extract of *S. officinalis* (100 and 200 mg/kg, p.o.) and its metabolites rosmarinic acid (10 and 20 mg/kg, i.p.) and caffeic acid (30 and 40 mg/kg, i.p.) produced a significant improvement in mechanical and thermal hyperalgesia and cold allodynia, and increased the sciatic functional index in mice [72]. The antinociceptive effects of the EtOAc extract of *S. amarissima* obtained in this study were greater than those observed in comparison with the reference drug tramadol, which is an atypical opioid analgesic that in addition to activating opioid receptors, increases monoamine concentrations as the mechanisms of its analgesic activity [73]. It is suggested that this atypical pathway of tramadol in the stimulation of monoamines favors the relief of FM-type pain [74]. Regarding amarisolide A (10 to 300 mg/kg), a dose-dependent response inhibition was observed for muscle hyperalgesia. This recovery was maintained throughout the 4 h of the test, unlike tramadol, which lost its significant effect after 150 min. The behavioral antinociceptive response of amarisolide A was similar to TR in the thermal and tactile allodynia test, where the aforementioned doses produced significant differences concerning the FM group. The temporal course curves showed that this recovery of the thresholds was maintained similarly throughout the entire test. Taken together, these results support the potential of *Salvia* species and their f terpenenic metabolites in relieving pain and being relevant for therapy, hitherto absent of FM-type pain.

Regarding the evaluation of the mechanisms of action for the antihyperalgesic and antiallodynic effects of amarisolide A in the reserpine-induced FM model, it was evidenced for the first time that pretreatment with the antagonist WAY100635 significantly reduced the analgesic effect of amarisolide A, allowing us to conclude that serotonin 5-HT_1A_ receptors are partially responsible for the antihyperalgesic and antiallodynic effects of amarisolide A and might be an interesting target for FM therapy. The in silico analysis in this investigation emphasized the possibility of a direct interaction of amarisolide A at the serotonin 5-HT_1A_ receptor to produce an agonistic response similar to buspirone. Activation of this receptor has been implicated in the alleviation of allodynia, hyperalgesia, and depressive-like behaviors [75]. The increase in swimming behavior in the FST would also support the idea that amarisolide A could activate the 5HT_1A_ receptor since an increase in such behavior has been correlated with antidepressant-like effects that can modulate the serotonergic activity [49,76,77]. Other studies have shown that activation of the 5-HT_1A_ receptor by the agonist NLX-112 can produce antihyperalgesic and antiallodynic effects in neuropathic pain [78]. It has also been shown that other natural products such as the mixture of diterpenes salvinorin A and B (30 mg/kg) and apigenin (3–30 mg/kg) can produce an analgesic effect by activating the serotonin 5-HT_1A_ receptor in formalin and the chronic constriction injury tests, respectively [24,79]. There are also reports that other *Salvia* species can produce an antidepressant-like effect. For example, the hydroalcoholic extract of *Salvia elegans* (125 to 2000 mg/kg) produced a significant reduction in immobility behavior in the FST [80], and the ethanolic extract of *Salvia lachnostachys* (100 mg/kg) improved the parameters related to the clonidine model of depression.

On the other hand, cytokine activation and regulation are actions involved in a complex manner in a variety of pathological states, for example, sepsis, rheumatoid arthritis, ankylosing spondylitis, Crohn’s disease, multiple sclerosis, and skin diseases [81]. Therefore, it is hypothesized that cytokines may be involved both in the generation of pain and hyperalgesia in inflammatory and neuropathic conditions [82,83] as well as in patients with FM, who have reported some behavioral-like symptoms of disease induced by cytokines [84]. The involvement of cytokines has already been reported in the reserpine-induced FM model [85,86]. In the present study, a preliminary exploration of IL-1β was performed in the spinal nervous tissue of rats with FM that received amarisolide A, where a significant reduction of this inflammatory cytokine was observed. This result agrees with that reported for tanshinone, a diterpene that effectively repressed spinal microglia activation in a neuropathic pain model, where the expression of inflammatory cytokines such as TNF-α and IL-1β was decreased in the spinal cord [87]. It is known that the activation of the NLRP3 inflammasome causes an increase in IL-1β concentrations [85] and that natural products such as ferulic acid (30 mg/kg) had an inhibitory effect of the inflammasome by producing a decrease in IL-1β concentrations [88]. Other natural products with potential antioxidant and anti-inflammatory properties such as quercetin have dose-dependently inhibited secretion of IL-1β from the NLRP3 inflammasome [89]. The results of our study reinforce the changes in the tissue levels of IL-1β in rats with FM, as has been observed in FM patients [90], as other possible mechanisms of action implicated in the antinociceptive effects of *S. amarissima* are due in part to the presence of amarisolide A.

## 5. Conclusions

In conclusion, this research provides preclinical scientific evidence of the antihyperalgesic, antiallodynic, and anti-depressive effects of *S. amarissima* against fibromyalgia-type pain, partly due to the clerodane-type diterpene amarisolide A isolated from a medium polar extract involving more than one molecular target, such as serotonin 5-HT_1A_ receptor and IL-1β. To the best of our knowledge, 5-HT_1A_ receptor agonists producing analgesia have not yet been specifically reported in preclinical FM-type pain or clinical trials. However, we hope that the preclinical results described in this study will encourage the investigation of serotonergic mechanisms such as the 5-HT_1A_ receptor in search of therapy for FM-type pain. All these results together in our research support the potential of this medicinal plant of the *Salvia* genus and the relevance of clerodane-type diterpenes such as amarisolide A, both as possible natural alternatives for people with FM.

## Figures and Tables

**Figure 1 metabolites-13-00059-f001:**
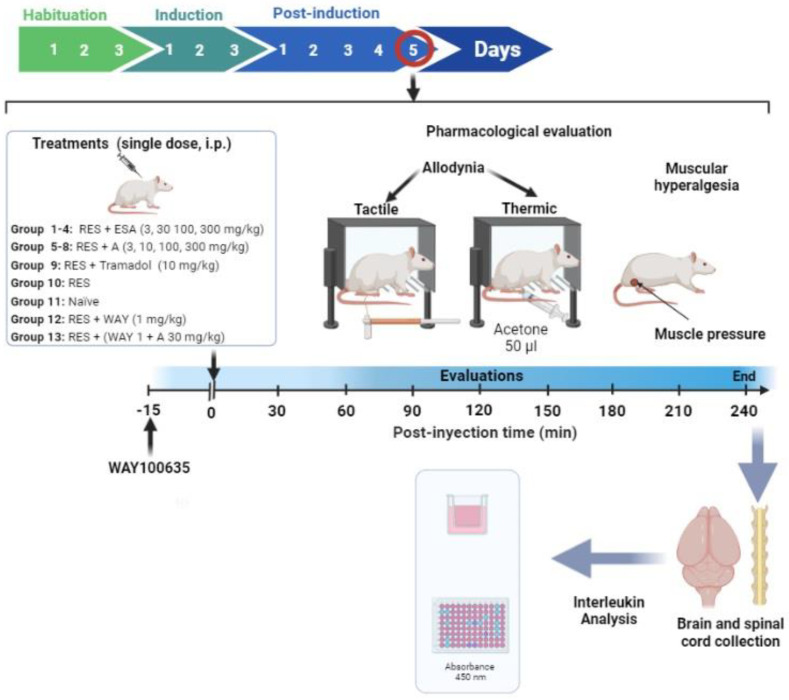
Timeline of the experimental design assessing the EtOAc extract of *Salvia amarissima* (ESA) and amarisolide A in the experimental FM-type pain induced by reserpine (RES).

**Figure 2 metabolites-13-00059-f002:**
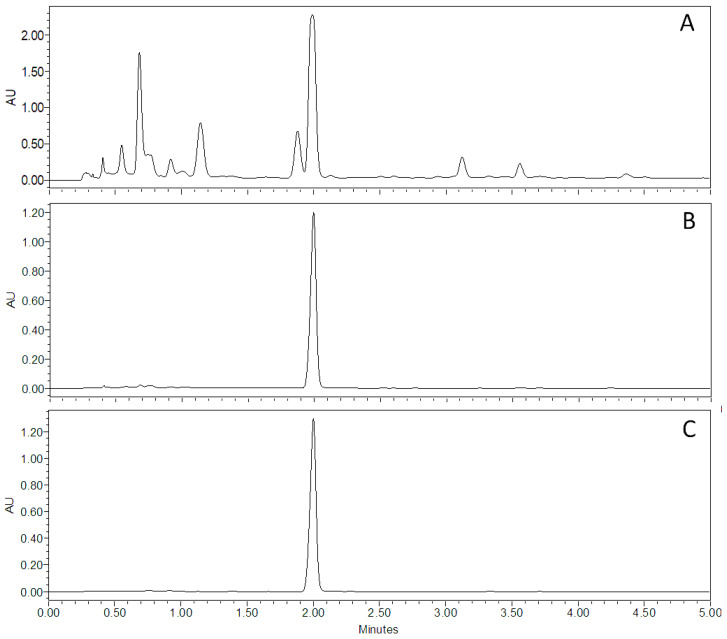
Chromatographic profile of the *S. amarissima* aerial parts in the (**A**) EtOAc extract, (**B**) amarisolide A isolated as a concentrated metabolite in the EtOAc extract, and (**C**) amarisolide A standard with a retention time of 1.99 min. All chromatograms were read at a wavelength of 256 nm.

**Figure 3 metabolites-13-00059-f003:**
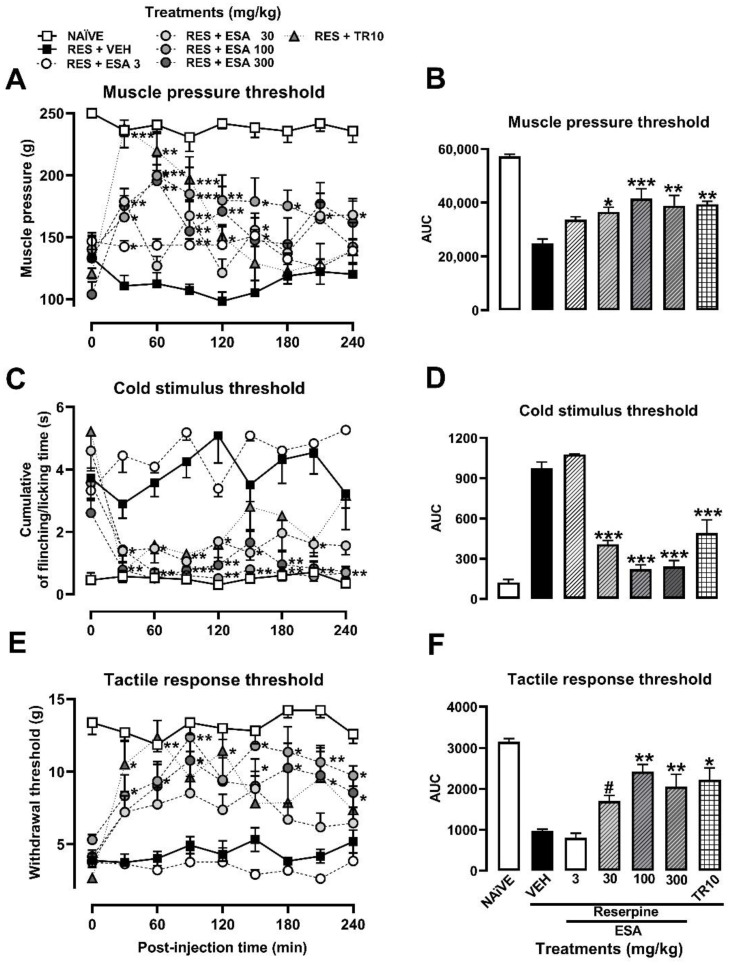
Antihyperalgesic and antiallodynic effect of *S. amarissima* EtOAc extract (ESA) on reserpine (RES)-induced fibromyalgia. (**A**) Muscle pressure threshold, (**C**) cold allodynia threshold, and (**E**) tactile response threshold on the 5th day post-induction. Each point represents the mean ± S.E.M of (**A**) the force in grams supported on the medial gastrocnemius muscle, (**C**) the accumulated response time to the cold stimulus (acetone) on the dorsal surface of the right hind limb for 1 min), (**E**) the force in grams supported on the plantar surface of the right hind limb (withdrawal threshold), compared to with the RES + VEH (reserpine, 1 mg/kg, s.c.), naïve and tramadol (TR10 mg/kg, i.p.) groups. Two-way repeated measures ANOVA followed by Dunnett’s post hoc test. * *p* < 0.05, ** *p* < 0.01, *** *p* < 0.001. Representation of the AUC of the time course obtained for (**B**) muscle pressure, (**D**) cold allodynia threshold, and (**F**) tactile response threshold. One-way ANOVA followed by Tukey’s post hoc test. * *p* < 0.05, ** *p* < 0.01, and *** *p* < 0.001 vs. the RES + VEH group. Student’s *t* test. ^#^ *p* < 0.05 vs. the RES + VEH group.

**Figure 4 metabolites-13-00059-f004:**
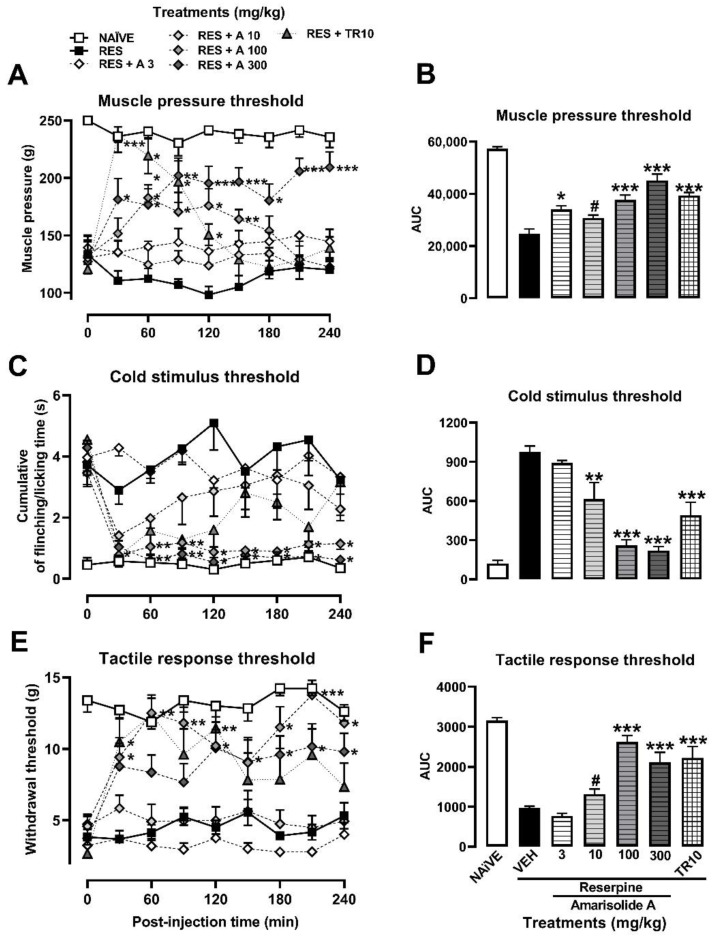
Antihyperalgesic and antiallodynic effects of amarisolide A on the reserpine (RES)-induced FM model. (**A**) Time course of the muscle pressure threshold. Each point represents the average mean ± S.E.M of the force in grams supported in the gastrocnemius muscle. (**B**) The area under the curve (AUC) of the time course obtained from each one of the treatments at the muscle pressure threshold. (**C**) Time course of cold allodynia threshold. Each point represents the mean ± S.E.M of the accumulated response time to the cold stimulus (acetone) on the dorsal surface of the right hind limb for 1 min. (**D**) The AUC at cold allodynia threshold. (**E**) Time course of tactile response threshold. Each point represents the mean ± S.E.M of the force in grams supported on the plantar surface of the right hind limb (withdrawal threshold). (**F**) The AUC at tactile response threshold. Two-way repeated measures ANOVA followed by Dunnett’s post hoc test (**A**,**C**,**E**) or one-way ANOVA followed by Tukey’s post hoc test (**B**,**D**,**F**). * *p* < 0.05, ** *p* < 0.01, *** *p* < 0.001, Student’s *t* test # *p* < 0.05 vs. RES + VEH group.

**Figure 5 metabolites-13-00059-f005:**
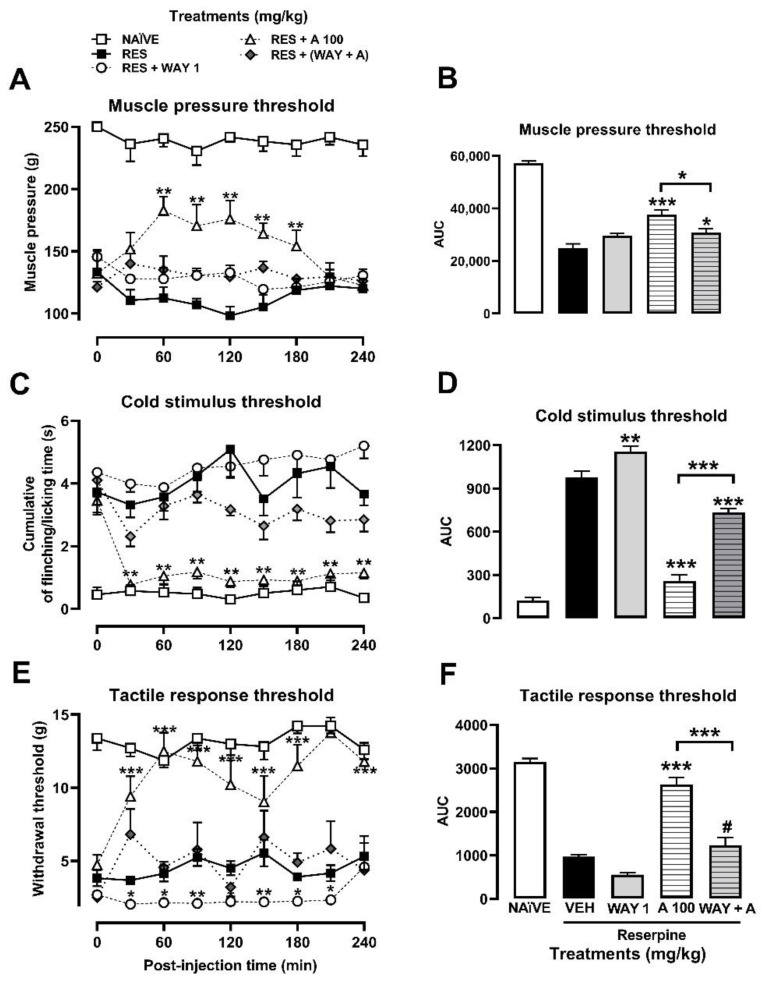
Evaluation of the serotonin 5-HT_1A_ receptor in the antihyperalgesic and antiallodynic effects of amarisolide A on the reserpine (RES)-induced FM model. (**A**) Time course of the muscle pressure threshold. Each point represents the mean ± S.E.M of the force in grams supported on the medial gastrocnemius muscle. (**B**) The AUC of the time course obtained from each one of the treatments at the muscle pressure threshold. (**C**) Time course of cold allodynia threshold. Each point represents the mean ± S.E.M of the accumulated response time to shaking or licking the right hind limb for 1 min. (**D**) The AUC at cold allodynia threshold. (**E**) Time course of tactile response threshold. Each point represents the mean ± S.E.M of the force in grams supported on the plantar surface of the right hind limb (withdrawal threshold). (**F**) The AUC at tactile response threshold. Two-way repeated measures ANOVA followed by the Dunnett’s post hoc test (**A**,**C**,**E**) or one-way ANOVA followed by Tukey’s post hoc test (**B**,**D**,**F**). * *p* < 0.05, ** *p* < 0.01, *** *p* < 0.001, Student’s *t* test ^#^ *p* < 0.05 vs. RES + VEH group. WAY 1 (WAY-100635, 1 mg/kg); A100 (amarisolide A, 100 mg/kg) and WAY + A (WAY-100635 1 mg/kg + amarisolide A 100 mg/kg).

**Figure 6 metabolites-13-00059-f006:**
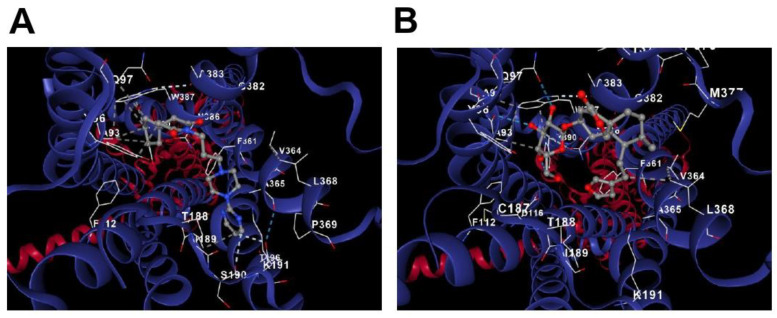
In silico evaluation of the interaction of buspirone or amarisolide A in the 5-HT_1A_ receptor. (**A**) Buspirone and 5-HT_1A_ receptor. (**B**) Amarisolide A and 5-HT_1A_ receptor. It is observed that in the amarisolide A-5-HT_1A_ complex, 11 steric interactions were shared in comparison with buspirone–5-HT_1A_ complex.

**Figure 7 metabolites-13-00059-f007:**
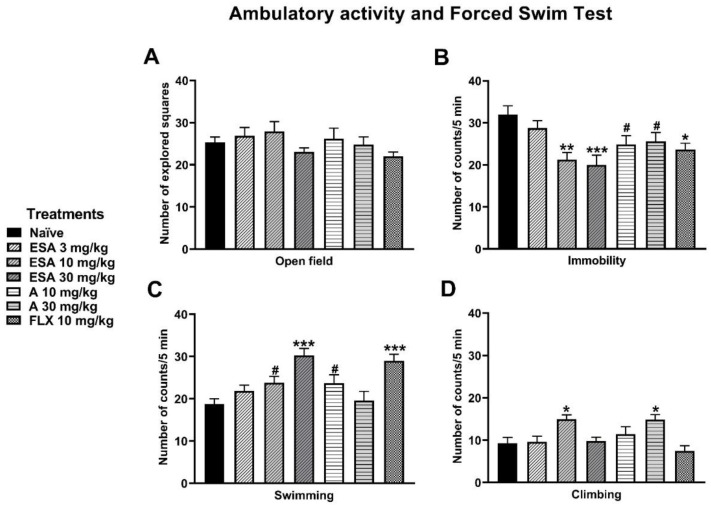
Effects of ESA and amarisolide A in the (**A**) ambulatory activity and (**B**–**D**) forced swimming test in rats. Data represent the mean ± S.E.M. One-way ANOVA followed by Tukey’s post hoc test.* *p* < 0.05, ** *p* < 0.01, *** *p* < 0.001 vs. naïve group. Unpaired Student’s *t* test ^#^ *p* < 0.05 vs. naïve group.

**Table 1 metabolites-13-00059-t001:** IL-1β levels expressed in pg/mg of nervous tissue from rats with and without fibromyalgia and in the presence of amarisolide A.

Tissue	Naïve	RES	RES + Amarisolide A (100 mg/kg)
**Prefrontal cortex**	0.0917 ± 0.1741	0.3820 ± 0.0153 &	0.3251 ± 0.005
**Hippocampus**	0.2507 ± 0.0758	0.4852 ± 0.005 &	0.0430 ± 0.002 ***
**Spinal cord:**			
**Thoraco-dorsal**	0.2806 ± 0.0739	0.5527 ± 0.2550 &	0.4322 ± 0.1528
**Lumbar-dorsal**	0.4293 ± 0.1500	1.255 ± 0.1931 &	0.4795 ± 0.1055 *
**Thoraco-ventral**	0.7343 ± 0.2312	0.9130 ± 0.1841	0.1895 ± 0.0745 *
**Lumbar-ventral**	0.1841 ± 0.1063	0.3110 ± 0.0929	0.0868 ± 0.0100 *

One-way ANOVA followed by the Tukey’s post hoc test. ^&^ *p* < 0.05 versus naïve group. * *p* < 0.05 and *** *p* < 0.001 versus reserpine (RES) group.

## Data Availability

The data presented in this study are available on request from the corresponding author due to privacy or ethical restrictions.

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
