# Peer review of "Antihyperalgesic and Antiallodynic Effects of Amarisolide A and Salvia amarissima Ortega in Experimental Fibromyalgia-Type Pain"

_metabolites, 2022, doi:10.3390/metabo13010059_

Round 1
Reviewer 1 Report
The manuscript “Antihyperalgesic and antiallodynic effects of Amarisolide A and Salvia amarissima Ortega in the experimental fibromyalgia-type pain” fits the journal’s scope. The authors present the non-clinical evaluation of an extract obtained from Salvia amarissima and its main compound amarisolide A, using the reserpine induced fibromyalgia-type chronic pain and in depressive-like behavior model. The effects evaluated were: antihyperalgesic, antiallodynic and antidepressive effects. The potential mechanism of action was also evaluated using a specific antagonist of 5-HT1A serotonin receptors. The aim of the research is clearly stated. Although some of the experiments are described in sufficient detail, the overall research design needs clarifications:
Animal experiments
1. Thus, the treatments and number of groups should be better presented. It is unclear if the treatment was administered in single dose, or over a period of time in the post-induction phase. The antidepressant activity of the EtOAc extract and amarisolide A should be further justify, mainly because of the treatment (single dose???).
2. Section 2.5.1. - please present clearly the groups and duration of treatment
3. Why were used only 6 animals/group, and not 8 (which is the minimum number accepted by the international guidelines)? The small number of animals corroborated with the high variability usually obtained in antinociceptive effect evaluation should be reflected in the SEM values. Please clarify the issue.
Extract preparation
1. Please indicate the extract quantities obtained. (from 50 g of plant material were obtained 25 g of EtOAc extract????)
2. From 25 g of EtOAc extract how much amarisolide A was obtained?
3. What aqueous extract? (lines 118-119)
4. The purification of amarisolide A is missing. The yield of 5.03 is from EtOAc extract, or it is reported to the plant material?
5. A reference standard of amarisolide A was used, or only the isolated compound?
Other remarks
Please add the number of the protocol for this research (not the general national guidelines)
Line 252 – please use the concentration, and not the volume
Please add the reference for the HPLC method
In the present form, the manuscript is not suitable for publication.
Author Response
December 13th, 2022
DEAR PROFESSOR DR KIM YANG
ASSOCIATED EDITOR
METABOLITES
Thank you for your e-mail dated December 7th about our submitted manuscript No. 2090073 Antihyperalgesic and antiallodynic effects of amarisolide A and Salvia amarissima Ortega in the experimental fibromyalgia-type pain to be considered for publishing in Metabolites. We thank all the suggestions and comments of the reviewers, we have included all of them to improve the manuscript.
I am submitting a revised version of the manuscript and a supplementary material according suggestions, comments, and questions. Our answers are appended below, and we hope you find this version suitable for publication.
Sincerely yours,
María Eva González-Trujano evag@imp.edu.mx
Laboratorio de Neurofarmacología de Productos Naturales
Dirección de Investigaciones en Neurociencias
Instituto Nacional de Psiquiatría Ramón de la Fuente Muñiz
Calz. México-Xochimilco No. 101, Col. Sn Lorenzo Huipulco
Delegación Tlalpan, Ciudad de México, 14370, México.
Phone: +(5255) 4160-5084
Reviewer 1
The manuscript “Antihyperalgesic and antiallodynic effects of Amarisolide A and Salvia amarissima Ortega in the experimental fibromyalgia-type pain” fits the journal’s scope. The authors present the non-clinical evaluation of an extract obtained from Salvia amarissima and its main compound amarisolide A, using the reserpine induced fibromyalgia-type chronic pain and in depressive-like behavior model. The effects evaluated were: antihyperalgesic, antiallodynic and antidepressive effects. The potential mechanism of action was also evaluated using a specific antagonist of 5-HT1A serotonin receptors. The aim of the research is clearly stated. Although some of the experiments are described in sufficient detail, the overall research design needs clarifications:
Animal experiments
- Thus, the treatments and number of groups should be better presented. It is unclear if the treatment was administered in single dose, or over a period of time in the post-induction phase. The antidepressant activity of the EtOAc extract and amarisolide A should be further justify, mainly because of the treatment (single dose???).
ANSWER: Thank you for your comments to clarify details in our experimental design and to improve this manuscript. Changes are indicated in the pdf document.
-Individual groups and duration of the treatment was clarified in the text and in the timeline included in figure 1. Point 2.5.1. Experimental design, lines 189-204 and 212-214.
-Since a maximum decrease or increase in the nociceptive thresholds is reached on the fifth day post-induction [22,33], the effects of acute treatment (single dose) were explored on that day. For this, repeated measures were evaluated in a period of 4 h to build temporal course curves later converted to dose-response plots using the area under the curve to determine their efficacy and potency parameters, and their tolerated doses. This information was added in the manuscript and in the timeline (Fig.1). Point 2.5.1. Experimental design, lines 189-204.
-Regarding the antidepressant activity, it is known that depression is the nearer comorbidity of FM-type pain and antidepressant drugs are the most effective therapy used currently. Because of this, the antidepressant activity of the EtOAc extract (3, 10, and 30 mg/kg) and amarisolide A (10 and 30 mg/kg) was investigated in this study compared to the antidepressant effects of the reference drug fluoxetine using three typical doses already reported in the literature [36,37]. Information included in point 2.5.1. Experimental design. 3rd paragraph, lines 210-215.
- Section
2.5.1. - please present clearly the groups and duration of treatment
ANSWER: Individual groups and duration of the treatment was clarified in the text and in the timeline included in figure 1. Point 2.5.1. Experimental design, lines 189-204 and 210-214.
- Why were used only 6 animals/group, and not 8 (which is the minimum number accepted by the international guidelines)? The small number of animals corroborated with the high variability usually obtained in antinociceptive effect evaluation should be reflected in the SEM values. Please clarify the issue.
ANSWER: We agree that variability in a nociplastic pain might be higher among the experimental individuals. Because of this, we included a n>8 in all the treatment groups receiving extracts, bioactive metabolite, or in the exploration of the mechanism of action. Whereas, groups with n=6 were used in the case of controls such as naïve and tramadol groups, since these groups have been explored and reported in previous studies in our laboratory. In addition, the statistical power test stated a power greater than 80% for all the cases. Number of animals per group was indicated in the text of point 2.5.1. Experimental design, line 189 and in 2.6 Statistical analysis (lines 335-336).
Extract preparation
- Please indicate the extract quantities obtained. (from 50 g of plant material were obtained 25 g of EtOAc extract????)
ANSWER: It was included in the text the amount obtained in grams (1.2575 g) and the percentage (5.03%) to clarify yielding of amarisolide A from the EtOAc extract using open chromatographic column. Point 2.1 Plant material, lines 141-145.
- From 25 g of EtOAc extract how much amarisolide A was obtained?
ANSWER: It was clarified in the text that two batches of the ethyl acetate extract (EtOAc) were required in the study. The first one was obtained and processed to identify the presence of amarisolide A and for concentration comparison of this metabolite among the different polarity extracts. While a second batch of EtOAc extract was prepared to isolate the bioactive metabolite using conventional chromatographic techniques to purify a major amount for its individual pharmacological evaluation. Point 2.1 Plant material, lines 132-145.
- What aqueous extract? (lines 118-119)
ANSWER: It was justified in the text that an aqueous extract of Salvia was carried out to determine the presence of amarisolide A compared with that obtained in the organic extracts from different polarity. From this comparison, the EtOAc was determined as the extract containing a major concentration of the bioactive metabolite. This analysis allowed to select this extract for behavioral assessments and to purify the amount of amarisolide A for its individual pharmacological evaluation. Point 2.1 Plant material, lines 126-129.
- The purification of amarisolide A is missing. The yield of 5.03 is from EtOAc extract, or it is reported to the plant material?
ANSWER: Details of the extraction, isolation, and purification of amarisolide A, as well as the yield (1.2575 g, 5.03%) obtained from the EtOAc extract (25 g) of a second batch of vegetal material (540 g) was included in the text for more clarity. Point 2.1 Plant material, lines 133-145.
- A reference standard of Americold A was used, or only the isolated compound?
ANSWER: Since amarisolide A is not an available commercial compound, it was compared by HPLC analysis with that conserved from the preliminary isolation and characterization using spectroscopic and spectrometric techniques already reported in the literature [22,33]. Information included in point 2.1 Plant material, lines 141-145.
Other remarks
Please add the number of the protocol for this research (not the general national guidelines)
ANSWER: The research protocols approved by our institutional committees were included in the text as No. NC123280.0 and NC17073.0, as well as the bioethical committee No. CONBIOETICA-09-CEI-01-20170316. Line 171 in point 2.3 Animals and in line 694 in Institutional Review Board Statement.
Line 252 – please use the concentration, and not the volume
ANSWER: information of dose and its corresponding concentration and volume of administration were included in the text for better clarity. Point 2.5.5 Antidepressant activity of the … 3rd paragraph, lines 310-311.
Please add the reference for the HPLC method
R: A reference for the HPLC method was included in the text as number 33, line 160, corresponding to:
- Moreno-Pérez et al., 2021 doi:10.1016/j.jep.2020.113550.
Reviewer 2
Authors investigated the effect of Salvia amarissima extract and its metabolite on pain symptoms in an animal model of fibromyalgia (FM). They focused on the 5-HT1A receptor and IL-1β as the mechanism involved in their analgesic effects. Based on the results, they demonstrate that these natural products are potential therapeutic options for patients suffering from FM pain. There are several crucial points authors need to consider.
Readability of text:
Authors may want to make the entire manuscript to be carefully checked regarding English grammar and phrasing by an English proofreader.
ANSWER: Thank you for your comments to clarify information and details in our experimental design to improve this manuscript, changes are indicated in the pdf document. English grammar and writing were extensively revised by an English proofreader.
Introduction:
“Dysfunctional pain” seems to be an old way of classifying chronic pain. FM pain is currently involved in “nociplastic pain” which IASP introduced in addition to “nociceptive” and ”neuropathic” pain in 2017.
ANSWER: The term “nociplastic” suggested as a new category for FM-type pain was justified in the text (introduction section in lines 53-55) and modified in the entire document.
Authors describe about NSAIDs and opioids as reference drugs in spite of that they are not standard drugs for FM. Authors need to refer to clinical efficacy and mechanisms of standard drugs such as pregabalin (Ca2+ channel inhibitor) and duloxetine (SNRI) instead of NSAIDs and opioids.
ANSWER: Information of current pharmacological therapy for FM was modified in the text to emphasize groups of drugs with major efficacy to relief FM-type pain according to their mechanism of action. Section of introduction, 3rd paragraph from lines 70-74.
The use of tramadol as reference drug was also reinforced in the text of introduction in the paragraph lines 78-84.
Authors may want to explain why they focused on the 5-HT1A receptor as a possible mechanism of action of the natural products.
ANSWER: In our experience in the laboratory, the 5-HT1A serotonin receptor is one of the inhibitory mechanisms of pain participating in the antinociceptive activity of several natural products such as flavonoids and terpenes, not only in visceral, nociceptive, or inflammatory pain, but also in neuropathic pain as it is also reported in literature by several researching groups (references below have been cited in discussion section). Nevertheless, data exploring antihyperalgesic and antiallodinic effects in the case of natural products and experimental FM is scarce. Antinociceptive properties of Salvia amarissima and its bioactive metabolite in other experimental models suggest actions at central level with participation of inhibitory mechanisms, one of these interesting targets to explore mechanisms of natural products in FM-type pain different than opioid receptors is 5-HT1A serotonin receptor. We have included information in the text to justify this condition in the experimental design of this study. Information to justify 5-HT1A receptor pharmacological evaluation was included in point 2.5.3 Mechanism of action…lines 265-267 to reinforce discussion in lines 620-637.
Miwa et al., 2009. https://doi.org/10.1038/AJG.2009.427
Lyubashina et al., 2017. https://doi.org/10.1134/S0012496617020119
Sagalajev et al., 2017. https://doi.org/10.1152/JN.00836.2016
Tlacomulco-Flores et al., 2020. doi:10.1016/j.jep.2019.112276.
Hernandez-Leon et al., 2020. doi:10.3390/molecules25030675.
Methods:
Direct method (e.g., receptor binding assay) for investigating 5-HT1A receptor blockade are necessary in addition to in silico experiments.
ANSWER: The preliminary exploration of the participation of the 5-HT1A receptor by using a selective antagonist is a good strategy in searching for evidence of the mechanism of action for amarisolide A and predictive in silico analysis as described in point 3.3 and 3.4. We agree that it would be interesting to carry out binding assays and other molecular tests to corroborate not only the binding or affinity on this receptor, but also the biochemical machinery of transduction and other targets. However, it is a limiting condition for us to performance radioactivity studies at this moment for requiring specialized equipment as a radiation detector, as well as a license to manage radioactivity, and availability of radioactive ligands.
Authors may want to investigate the effect of 5-HT1A receptor agonist in the reserpine model if they consider that the stimulation of 5-HT1A receptor shows analgesia in the FM animal model.
ANSWER: We agree. In our acknowledge, the 5-HT1A receptor agonists producing analgesia are not yet specifically reported in the reserpine-induced model of FM-type pain. Nevertheless, since it was observed in this study that the presence of WAY100635 modified the antihyperalgesic and antiallodynic effects of amarisolide A, it suggests the participation of this receptor in the FM model and our data of prediction in in silico analysis using buspirone also suggested equivalent interactions with amarisolide A. It will be interesting in the future to analyze and compare the in vivo efficacy of amarisolide A and other agonist ligands on this inhibitory receptor in FM models, since it has already reported the antinociceptive activity of 8-OH-DPAT and buspirone as agonists of the 5-HT1A serotonin receptor in models of acute and chronic pain, as well as an antidepressant-like effect at a preclinical level in references cited below and included in the text of discussion.
Belcheva, S., Petkov, V.D., Konstantinova, E., Petkov, V. V., Boyanova, E., 1995. Effects on nociception of the Ca2+ and 5-HT antagonist dotarizine and other 5-HT receptor agonists and antagonists. Acta Physiol. Pharmacol. Bulg. 21, 93–98.
Galeotti, N., Ghelardini, C., Bartolini, A., 1997. 5-HT1A Agonists Induce Central Cholinergic Antinociception. Pharmacol. Biochem. Behav. 57, 835–841. https://doi.org/10.1016/S0091-3057(96)00401-7
Gjerstad, J., Tjølsen, A., Hole, K., 1996. The effect of 5-HT1A receptor stimulation on nociceptive dorsal horn neurones in rats. Eur. J. Pharmacol. 318, 315–321. https://doi.org/10.1016/S0014-2999(96)00819-9
Haleem, D.J., 2019. Targeting Serotonin1A Receptors for Treating Chronic Pain and Depression. Curr. Neuropharmacol. 17, 1098. https://doi.org/10.2174/1570159X17666190811161807
Mjellem, N., Lund, A., Eide, P.K., Sterkson, R., TjØlsen, A., 1992. The role of 5-HT1A and 5-HT1B receptors in spinal nociceptive transmission and in the modulation of NMDA induced behaviour. Neuroreport 3, 1061–1064. https://doi.org/10.1097/00001756-199212000-00007
Rojas, P.S., Fiedler, J.L., Bello, A., Wang, H., Ciranna, L., 2016. What Do We Really Know About 5-HT 1A Receptor Signaling in Neuronal Cells? https://doi.org/10.3389/fncel.2016.00272
Discussion:
Authors need to explain the reason why they used tramadol as the positive reference drug in spite of pregabalin and duloxetine. Tramadol does not seem a standard and effective analgesic in the treatment of FM pain.
ANSWER: Tramadol is a drug already used as monotherapy and/or in combination with an antidepressant or analgesic drugs with moderate efficacy in FM (Macfarlane et al., 2017). It is due to tramadol produces its analgesic effect not only by modulation of the μ receptor but also by inhibition of the serotonin and noradrenaline reuptake. Experimental studies reported that tramadol has been used in comparison to placebo in some clinical trials directed to patients with severe FM (Biasi et al., 1998; da Rocha et al., 2020), and it has already been reported as a reference drug in the experimental model of FM induced by reserpine (Kaneko et al., 2014, Quinto-Ortíz et al. 2022). For these reasons, in part, it was used as a reference drug in this study. Regarding other drugs used for FM-type pain, we have explored the effects of gabapentin in the reserpine-induced FM model. However, experimental conditions used in the evaluation of this drug were a little different to those used in this study, we have included in the supplementary material data not reported.
Biasi, G.; Manca, S.; Manganelli, S.; Marcolongo, R. Tramadol in the fibromyalgia syndrome: A controlled clinical trial versus placebo. Int. J. Clin. Pharmacol. Res. 1998, 18, 13–19.
da Rocha, A.P.; Mizzaci, C.C.; Pinto, A.C.P.N.; da Silva Vieira, A.G.; Civile, V.; Trevisani, V.F.M. Tramadol for management of fibromyalgia pain and symptoms: Systematic review. Int. J. Clin. Pract. 2020, 74, e13455
Kaneko, K.; Umehara, M.; Homan, T.; Okamoto, K.; Oka, M.; Oyama, T. The analgesic effect of tramadol in animal models of neuropathic pain and fibromyalgia. Neurosci. Lett. 2014, 562, 28–33.
Quinto-Ortiz, Y.E.; González-Trujano, M.E.; Sánchez-Jaramillo, E.; Moreno-Pérez, G.F.; Jacinto-Gutiérrez, S.; Pellicer, F.; Fernández-Guasti, A.; Hernandez-Leon, A. Pharmacological Interaction of Quercetin Derivatives of Tilia Americana and Clinical Drugs in Experimental Fibromyalgia. Metabolites 2022, 12, 916, doi:10.3390/METABO12100916/S1.
The analgesic efficacy of 5-HT1A receptor agonists (e.g., tandospirone) in the clinic can be cited, if any.
ANSWER: To our knowledge, 5-HT1A receptor agonists producing analgesia are not yet specifically reported in the reserpine-induced FM-type pain or in clinical trials like in other kind of pain (for example, buspirone is suggested to be efficacious in abdominal pain https://doi.org/10.14309/ajg.0000000000000589). However, we hope that the preclinical results described in this study encourage investigation of serotonergic mechanisms, specifically of 5-HT1A receptor in search of FM-type pain therapy. A phrase was included in point 5. Conclusions, lines 669-673.

Reviewer 2 Report
Authors investigated the effect of Salvia amarissima extract and its metabolite on pain symptoms in an animal model of fibromyalgia (FM). They focused on the 5-HT1A receptor and IL-1β as the mechanism involved in their analgesic effects. Based on the results, they demonstrate that these natural products are potential therapeutic options for patients suffering from FM pain. There are several crucial points authors need to consider.
Readability of text:
Authors may want to make the entire manuscript to be carefully checked regarding English grammar and phrasing by an English proofreader.
Introduction:
“Dysfunctional pain” seems to be an old way of classifying chronic pain. FM pain is currently involved in “nociplastic pain” which IASP introduced in addition to “nociceptive” and ”neuropathic” pain in 2017.
Authors describe about NSAIDs and opioids as reference drugs in spite of that they are not standard drugs for FM. Authors need to refer to clinical efficacy and mechanisms of standard drugs such as pregabalin (Ca2+ channel inhibitor) and duloxetine (SNRI) instead of NSAIDs and opioids.
Authors may want to explain why they focused on the 5-HT1A receptor as a possible mechanism of action of the natural products.
Methods:
Direct method (e.g., receptor binding assay) for investigating 5-HT1A receptor blockade are necessary in addition to in silico experiments.
Authors may want to investigate the effect of 5-HT1A receptor agonist in the reserpine model if they consider that the stimulation of 5-HT1A receptor shows analgesia in the FM animal model.
Discussion:
Authors need to explain the reason why they used tramadol as the positive reference drug in spite of pregabalin and duloxetine. Tramadol does not seem a standard and effective analgesic in the treatment of FM pain.
The analgesic efficacy of 5-HT1A receptor agonists (e.g., tandospirone) in the clinic can be cited, if any.
Author Response
Reviewer 2
Authors investigated the effect of Salvia amarissima extract and its metabolite on pain symptoms in an animal model of fibromyalgia (FM). They focused on the 5-HT1A receptor and IL-1β as the mechanism involved in their analgesic effects. Based on the results, they demonstrate that these natural products are potential therapeutic options for patients suffering from FM pain. There are several crucial points authors need to consider.
Readability of text:
Authors may want to make the entire manuscript to be carefully checked regarding English grammar and phrasing by an English proofreader.
ANSWER: Thank you for your comments to clarify information and details in our experimental design to improve this manuscript, changes are indicated in the pdf document. English grammar and writing were extensively revised by an English proofreader.
Introduction:
“Dysfunctional pain” seems to be an old way of classifying chronic pain. FM pain is currently involved in “nociplastic pain” which IASP introduced in addition to “nociceptive” and ”neuropathic” pain in 2017.
ANSWER: The term “nociplastic” suggested as a new category for FM-type pain was justified in the text (introduction section in lines 53-55) and modified in the entire document.
Authors describe about NSAIDs and opioids as reference drugs in spite of that they are not standard drugs for FM. Authors need to refer to clinical efficacy and mechanisms of standard drugs such as pregabalin (Ca2+ channel inhibitor) and duloxetine (SNRI) instead of NSAIDs and opioids.
ANSWER: Information of current pharmacological therapy for FM was modified in the text to emphasize groups of drugs with major efficacy to relief FM-type pain according to their mechanism of action. Section of introduction, 3rd paragraph from lines 70-74.
The use of tramadol as reference drug was also reinforced in the text of introduction in the paragraph lines 78-84.
Authors may want to explain why they focused on the 5-HT1A receptor as a possible mechanism of action of the natural products.
ANSWER: In our experience in the laboratory, the 5-HT1A serotonin receptor is one of the inhibitory mechanisms of pain participating in the antinociceptive activity of several natural products such as flavonoids and terpenes, not only in visceral, nociceptive, or inflammatory pain, but also in neuropathic pain as it is also reported in literature by several researching groups (references below have been cited in discussion section). Nevertheless, data exploring antihyperalgesic and antiallodinic effects in the case of natural products and experimental FM is scarce. Antinociceptive properties of Salvia amarissima and its bioactive metabolite in other experimental models suggest actions at central level with participation of inhibitory mechanisms, one of these interesting targets to explore mechanisms of natural products in FM-type pain different than opioid receptors is 5-HT1A serotonin receptor. We have included information in the text to justify this condition in the experimental design of this study. Information to justify 5-HT1A receptor pharmacological evaluation was included in point 2.5.3 Mechanism of action…lines 265-267 to reinforce discussion in lines 620-637.
Miwa et al., 2009. https://doi.org/10.1038/AJG.2009.427
Lyubashina et al., 2017. https://doi.org/10.1134/S0012496617020119
Sagalajev et al., 2017. https://doi.org/10.1152/JN.00836.2016
Tlacomulco-Flores et al., 2020. doi:10.1016/j.jep.2019.112276.
Hernandez-Leon et al., 2020. doi:10.3390/molecules25030675.
Methods:
Direct method (e.g., receptor binding assay) for investigating 5-HT1A receptor blockade are necessary in addition to in silico experiments.
ANSWER: The preliminary exploration of the participation of the 5-HT1A receptor by using a selective antagonist is a good strategy in searching for evidence of the mechanism of action for amarisolide A and predictive in silico analysis as described in point 3.3 and 3.4. We agree that it would be interesting to carry out binding assays and other molecular tests to corroborate not only the binding or affinity on this receptor, but also the biochemical machinery of transduction and other targets. However, it is a limiting condition for us to performance radioactivity studies at this moment for requiring specialized equipment as a radiation detector, as well as a license to manage radioactivity, and availability of radioactive ligands.
Authors may want to investigate the effect of 5-HT1A receptor agonist in the reserpine model if they consider that the stimulation of 5-HT1A receptor shows analgesia in the FM animal model.
ANSWER: We agree. In our acknowledge, the 5-HT1A receptor agonists producing analgesia are not yet specifically reported in the reserpine-induced model of FM-type pain. Nevertheless, since it was observed in this study that the presence of WAY100635 modified the antihyperalgesic and antiallodynic effects of amarisolide A, it suggests the participation of this receptor in the FM model and our data of prediction in in silico analysis using buspirone also suggested equivalent interactions with amarisolide A. It will be interesting in the future to analyze and compare the in vivo efficacy of amarisolide A and other agonist ligands on this inhibitory receptor in FM models, since it has already reported the antinociceptive activity of 8-OH-DPAT and buspirone as agonists of the 5-HT1A serotonin receptor in models of acute and chronic pain, as well as an antidepressant-like effect at a preclinical level in references cited below and included in the text of discussion.
Belcheva, S., Petkov, V.D., Konstantinova, E., Petkov, V. V., Boyanova, E., 1995. Effects on nociception of the Ca2+ and 5-HT antagonist dotarizine and other 5-HT receptor agonists and antagonists. Acta Physiol. Pharmacol. Bulg. 21, 93–98.
Galeotti, N., Ghelardini, C., Bartolini, A., 1997. 5-HT1A Agonists Induce Central Cholinergic Antinociception. Pharmacol. Biochem. Behav. 57, 835–841. https://doi.org/10.1016/S0091-3057(96)00401-7
Gjerstad, J., Tjølsen, A., Hole, K., 1996. The effect of 5-HT1A receptor stimulation on nociceptive dorsal horn neurones in rats. Eur. J. Pharmacol. 318, 315–321. https://doi.org/10.1016/S0014-2999(96)00819-9
Haleem, D.J., 2019. Targeting Serotonin1A Receptors for Treating Chronic Pain and Depression. Curr. Neuropharmacol. 17, 1098. https://doi.org/10.2174/1570159X17666190811161807
Mjellem, N., Lund, A., Eide, P.K., Sterkson, R., TjØlsen, A., 1992. The role of 5-HT1A and 5-HT1B receptors in spinal nociceptive transmission and in the modulation of NMDA induced behaviour. Neuroreport 3, 1061–1064. https://doi.org/10.1097/00001756-199212000-00007
Rojas, P.S., Fiedler, J.L., Bello, A., Wang, H., Ciranna, L., 2016. What Do We Really Know About 5-HT 1A Receptor Signaling in Neuronal Cells? https://doi.org/10.3389/fncel.2016.00272
Discussion:
Authors need to explain the reason why they used tramadol as the positive reference drug in spite of pregabalin and duloxetine. Tramadol does not seem a standard and effective analgesic in the treatment of FM pain.
ANSWER: Tramadol is a drug already used as monotherapy and/or in combination with an antidepressant or analgesic drugs with moderate efficacy in FM (Macfarlane et al., 2017). It is due to tramadol produces its analgesic effect not only by modulation of the μ receptor but also by inhibition of the serotonin and noradrenaline reuptake. Experimental studies reported that tramadol has been used in comparison to placebo in some clinical trials directed to patients with severe FM (Biasi et al., 1998; da Rocha et al., 2020), and it has already been reported as a reference drug in the experimental model of FM induced by reserpine (Kaneko et al., 2014, Quinto-Ortíz et al. 2022). For these reasons, in part, it was used as a reference drug in this study. Regarding other drugs used for FM-type pain, we have explored the effects of gabapentin in the reserpine-induced FM model. However, experimental conditions used in the evaluation of this drug were a little different to those used in this study, we have included in the supplementary material data not reported.
Biasi, G.; Manca, S.; Manganelli, S.; Marcolongo, R. Tramadol in the fibromyalgia syndrome: A controlled clinical trial versus placebo. Int. J. Clin. Pharmacol. Res. 1998, 18, 13–19.
da Rocha, A.P.; Mizzaci, C.C.; Pinto, A.C.P.N.; da Silva Vieira, A.G.; Civile, V.; Trevisani, V.F.M. Tramadol for management of fibromyalgia pain and symptoms: Systematic review. Int. J. Clin. Pract. 2020, 74, e13455
Kaneko, K.; Umehara, M.; Homan, T.; Okamoto, K.; Oka, M.; Oyama, T. The analgesic effect of tramadol in animal models of neuropathic pain and fibromyalgia. Neurosci. Lett. 2014, 562, 28–33.
Quinto-Ortiz, Y.E.; González-Trujano, M.E.; Sánchez-Jaramillo, E.; Moreno-Pérez, G.F.; Jacinto-Gutiérrez, S.; Pellicer, F.; Fernández-Guasti, A.; Hernandez-Leon, A. Pharmacological Interaction of Quercetin Derivatives of Tilia Americana and Clinical Drugs in Experimental Fibromyalgia. Metabolites 2022, 12, 916, doi:10.3390/METABO12100916/S1.
The analgesic efficacy of 5-HT1A receptor agonists (e.g., tandospirone) in the clinic can be cited, if any.
ANSWER: To our knowledge, 5-HT1A receptor agonists producing analgesia are not yet specifically reported in the reserpine-induced FM-type pain or in clinical trials like in other kind of pain (for example, buspirone is suggested to be efficacious in abdominal pain https://doi.org/10.14309/ajg.0000000000000589). However, we hope that the preclinical results described in this study encourage investigation of serotonergic mechanisms, specifically of 5-HT1A receptor in search of FM-type pain therapy. A phrase was included in point 5. Conclusions, lines 669-673.

Round 2
Reviewer 1 Report
The authors clarified / corrected the raised issues. In the present form, the manuscript is suitable for publication.
Author Response
Reviewer 1
The authors clarified / corrected the raised issues. In the present form, the manuscript is suitable for publication
ANSWER: Thank you very much, we appreciate your time and comments to improve our manuscript.
Reviewer 2 Report
There are still several crucial points authors need to consider.
Readability of text:
The proofread seems far from adequate. It seems the author's responsibility to use standard English grammar. There are many sentences which seem to need a correction. The followings are just examples to be considered. There seems to be no distinction between adjectives and adverbs, or verbs and nouns for authors.
L27: To get scientific evidence of its potential effects to relief (→ relieve) nociplastic pain…
L39: In part because of the presence of amarisolide A involving the 5-HT1A serotonin receptor. (→ this does not form a sentence).
L109: As well as the antidepressant activity (→ antidepressant activity was also investigated) since depression is a closely (→ close) comorbidity of FM pain.
The sentences above are just examples.
Additional experiments
It is not appropriate to claim an action on the 5-HT1A receptor based on in silico calculation. Methods other than binding assay is available (e.g. cAMP assay) for demonstrating effects on the 5-HT receptor. Authors may want to consider such experiments to prove the mechanism action.
Authors may want to investigate 5-HT1A receptor agonists (e.g., buspirone) in RIM reserpine (FM) model for demonstrating therapeutic potential of 5-HT1A stimulation in treating FM pain. Because the pathophysiology of FM seems different from other types of pain, results in animal models of other types of pain (references authors raised) are not so helpful.
Author Response
Reviewer 2
There are still several crucial points authors need to consider.
Readability of text:
The proofread seems far from adequate. It seems the author's responsibility to use standard English grammar. There are many sentences which seem to need a correction. The followings are just examples to be considered. There seems to be no distinction between adjectives and adverbs, or verbs and nouns for authors.
L27: To get scientific evidence of its potential effects to relief (→ relieve) nociplastic pain…
L39: In part because of the presence of amarisolide A involving the 5-HT1A serotonin receptor. (→ this does not form a sentence).
L109: As well as the antidepressant activity (→ antidepressant activity was also investigated) since depression is a closely (→ close) comorbidity of FM pain.
The sentences above are just examples.
ANSWER: Thank you so much for your time. My apologies for the misspellings, track changes make it hard to spot them sometimes. We are sending a new proofreading version.
Additional experiments
It is not appropriate to claim an action on the 5-HT1A receptor based on in silico calculation. Methods other than binding assay is available (e.g. cAMP assay) for demonstrating effects on the 5-HT receptor. Authors may want to consider such experiments to prove the mechanism action.
ANSWER: We agree to this comment. It is important to mention that our present results are part of the experiments included in the Fernando´s PhD thesis. In the thesis, the antinociceptive effects of amarisolide A were also explored in the neurogenic and inflammatory phases of the formalin test in mice. In addition, the 5-HT1A serotonin receptor and the cAMP involvement (as a signal pathway of receptors coupled to the Gi protein) were evaluated to obtain preliminary data of the possible mechanism of action. For this, it was used forskolin (10 mg/kg, i.p.) to activate adenylylcyclase and theophylline (20 mg/kg, i.p.) to prevent the intracellular break-down of cAMP. The following figure shows that pretreatment with forskolin and theophylline inhibited the significant antinociceptive response of amarisolide A in both phases, the neurogenic and inflammatory, in the formalin test in mice. These data were the antecedent to explore the 5-HT1A serotonin receptor involvement in the presence of amarisolide A in the experimental FM-type pain submitted to metabolites journal.
Figure included in the attached file.
Legend of the figure. Antinociceptive effects of amarisolide A (1 mg/kg) expressed as neurogenic (AUC0-10 min) or inflammatory (AUC10-30 min) phase in the 1% formalin test in mice receiving: vehicle (VEH), amarisolide A, the 5-HT1A serotonin receptor antagonist WAY100365 (WAY, 1 mg/kg), the adenylyl cyclase activator forskolin (FSK, 10 mg/kg), or the phosphodiesterase inhibitor theophylline (THP, 20 mg/kg), alone or combined. Data are expressed as the mean ± S.E.M. of at least six repetitions. One-way ANOVA followed by Dunnett’s post-hoc test. Neurogenic phase F4,29=5.22, p=0.0027 and inflammatory phase F4,29=50.24, p=0.0001. *p<0.05, ***p < 0.001, ****p<0.0001.
Authors may want to investigate 5-HT1A receptor agonists (e.g., buspirone) in RIM reserpine (FM) model for demonstrating therapeutic potential of 5-HT1A stimulation in treating FM pain. Because the pathophysiology of FM seems different from other types of pain, results in animal models of other types of pain (references authors raised) are not so helpful.
ANSWER: We agree to your comment that pathophysiology of FM shows differences in comparison to other types of pain. To the best of our knowledge, this is the first study reporting participation of the 5-HT1A receptor in FM-type pain and in the presence of a natural product. Since, this receptor is coupled to cAMP pathway and activation of the cAMP pathway in the central nervous system is considered a mediator producing hyperalgesia and inhibition of this pathway reduces hyperalgesia in animal models of acute and chronic pain. It will be interesting to continue the study of this mechanism in more detail, not only in the involvement of the 5-HT1A serotonin receptor and cAMP pathway in the presence of amarisolide A but also by using selective agonists and other pharmacological tools for better comprehension of its relevance in the FM therapy.

Round 3
Reviewer 2 Report
It becomes easy to read the revised manuscript because English has been improved.